# Whole Breast Irradiation in Comparison to Endocrine Therapy in Early Stage Breast Cancer—A Direct and Network Meta-Analysis of Published Randomized Trials

**DOI:** 10.3390/cancers15174343

**Published:** 2023-08-30

**Authors:** Jan Haussmann, Wilfried Budach, Stefanie Corradini, David Krug, Edwin Bölke, Balint Tamaskovics, Danny Jazmati, Alexander Haussmann, Christiane Matuschek

**Affiliations:** 1Department of Radiation Oncology, Medical Faculty and University Hospital Düsseldorf, Heinrich Heine University, 40225 Düsseldorf, Germany; jan.haussmann@med.uni-duesseldorf.de (J.H.); wilfried.budach@med.uni-duesseldorf.de (W.B.); balint.tamaskovics@med.uni-duesseldorf.de (B.T.); danny.jazmati@med.uni-duesseldorf.de (D.J.); matuschek@med.uni-duesseldorf.de (C.M.); 2Department of Radiation Oncology, University Hospital, Ludwig-Maximilians-University (LMU), 81377 Munich, Germany; stefanie.corradini@med.uni-muenchen.de; 3Department of Radiation Oncology, University Hospital Schleswig-Holstein, 24105 Kiel, Germany; david.krug@uksh.de; 4Division of Physical Activity, Prevention and Cancer, German Cancer Research Center (DKFZ), 69120 Heidelberg, Germany; alexander.haussmann@nct-heidelberg.de

**Keywords:** Network meta-analysis, breast cancer, radiotherapy, endocrine therapy, de-escalation

## Abstract

**Simple Summary:**

In order to avoid side effects from treatment, patients suffering from breast cancer with a lower risk of relapse might forgo radiation therapy to the whole breast or endocrine therapy after surgery. In this analysis, we compared these two options regarding the risk of breast cancer relapse with the help of direct trials and a network that analyzed one of the two options. We found that both treatment options have similar long-term cancer outcomes and should be considered equally effective.

**Abstract:**

Background: Multiple randomized trials have established adjuvant endocrine therapy (ET) and whole breast irradiation (WBI) as the standard approach after breast-conserving surgery (BCS) in early-stage breast cancer. The omission of WBI has been studied in multiple trials and resulted in reduced local control with maintained survival rates and has therefore been adapted as a treatment option in selected patients in several guidelines. Omitting ET instead of WBI might also be a valuable option as both treatments have distinctly different side effect profiles. However, the clinical outcomes of BCS + ET vs. BCS + WBI have not been formally analyzed. Methods: We performed a systematic literature review searching for randomized trials comparing BCS + ET vs. BCS + WBI in low-risk breast cancer patients with publication dates after 2000. We excluded trials using any form of chemotherapy, regional nodal radiation and mastectomy. The meta-analysis was performed using a two-step process. First, we extracted all available published event rates and the effect sizes for overall and breast-cancer-specific survival (OS, BCSS), local (LR) and regional recurrence, disease-free survival, distant metastases-free interval, contralateral breast cancer, second cancer other than breast cancer and mastectomy-free interval as investigated endpoints and compared them in a network meta-analysis. Second, the published individual patient data from the Early Breast Cancer Trialists’ Collaborative Group (EBCTCG) publications were used to allow a comparison of OS and BCSS. Results: We identified three studies, including a direct comparison of BCS + ET vs. BCS + WBI (*n* = 1059) and nine studies randomizing overall 7207 patients additionally to BCS only and BCS + WBI + ET resulting in a four-arm comparison. In the network analysis, LR was significantly lower in the BCS + WBI group in comparison with the BCS + ET group (HR = 0.62; CI-95%: 0.42–0.92; *p* = 0.019). We did not find any differences in OS (HR = 0.93; CI-95%: 0.53–1.62; *p* = 0.785) and BCSS (OR = 1.04; CI-95%: 0.45–2.41; *p* = 0.928). Further, we found a lower distant metastasis-free interval, a higher rate of contralateral breast cancer and a reduced mastectomy-free interval in the BCS + WBI-arm. Using the EBCTCG data, OS and BCSS were not significantly different between BCS + ET and BCS + WBI after 10 years (OS: OR = 0.85; CI-95%: 0.59–1.22; *p* = 0.369) (BCSS: OR = 0.72; CI-95%: 0.38–1.36; *p* = 0.305). Conclusion: Evidence from direct and indirect comparison suggests that BCS + WBI might be an equivalent de-escalation strategy to BCS + ET in low-risk breast cancer. Adverse events and quality of life measures have to be further compared between these approaches.

## 1. Introduction

Multiple randomized trials have established breast-conserving surgery (BCS), adjuvant systemic therapy and whole breast irradiation (WBI) as the standard in early-stage breast cancer treatment. A meta-analysis by The Early Breast Cancer Trialists’ Collaborative Group (EBCTCG) provided robust evidence that adjuvant WBI after breast-conserving surgery improves local control and overall survival [1]. A subsequent analysis by the same group also showed that the addition of tamoxifen reduces mortality compared with no endocrine treatment leading to the current standard of care [2]. The use of aromatase inhibitors (AI) instead of tamoxifen further improved outcomes [3].

While achieving gratifying oncological results with this approach, recent efforts have focused on treatment de-intensification in presumed low-risk patients (i.e., small primary tumors, low or intermediate grading, low proliferation index, hormone receptor-positive cancers).

One suggested option for treatment de-intensification might be to omit radiation therapy. This was put forward in order to allow an omission of the seldom, but possibly debilitating, long-term side effects of radiation therapy, which can include arm and breast symptoms, breast tissue fibrosis, lung and heart toxicity, as well as second malignancies of the contralateral breast, lung and the irradiated skin [4]. Further, the EBCTCG analysis also demonstrated that, despite a consistent relative benefit, the absolute benefit of WBI in any first recurrence (absolute benefit ~5% after 10 years) and survival is very small in elderly women with hormone receptor-positive breast cancer [1].

Adjuvant endocrine therapy (ET) has been discussed in the literature as a favorable treatment option compared with adjuvant WBI. Several trials that randomized patients to adjuvant endocrine therapy with or without WBI [5,6,7,8,9,10] found no significant differences in overall survival. However, meta-analyses found that local control is inferior when radiation therapy is omitted [11,12]. So far, attempts to identify a subgroup without a benefit from adjuvant radiotherapy have not been successful [13], but further prospective trials incorporating gene expression analysis are ongoing [14,15,16,17,18,19].

As part of a treatment de-intensification, one could also consider the omission of a long-term ET and confining adjuvant treatment to whole breast radiation alone. This would avoid debilitation side effects of ET, such as arthralgia, osteopenia as well as vaginal dryness, and could allow patients to benefit from specific advantages of WBI (e.g., increased local control). The present paper addresses the comparison of both treatments in randomized trials.

## 2. Material and Methods

We conducted a systematic literature search of the electronic database PubMed for randomized controlled trials comparing adjuvant endocrine therapy to radiation therapy in breast cancer after breast-conserving surgery in accordance with the published PRISMA guidelines [20] on 4 April 2023. The used search words were “(radiotherapy OR radiation OR irradiation) AND (endocrine OR tamoxifen OR aromatase inhibitor) AND (“breast cancer” OR “adenocarcinoma breast”) AND (randomized OR randomised OR randomly)”. Further, we screened the major scientific meetings (e.g., ASCO, ASTRO, ESMO, ESTRO, AACR annual meetings) with the same keywords for published abstracts.

We included randomized trials for early-stage breast cancer, comparing any type of ET to WBI. In order to minimize heterogeneity and maximize the homogeneity of the compared study populations, only low-risk populations were included. Eligibility criteria included T-Stage T1-2, node-negative disease and breast-conserving surgery. We excluded trials using mastectomy and preoperative or adjuvant chemotherapy. We also excluded trials that used regional nodal irradiation as we consider these patients to be at a higher risk for local and distant recurrence. All studies had to have published 5-year results after 1 January 2000.

In order to expand the analysis using a direct as well as an indirect comparison, we also searched for trials with the same inclusion criteria treating patients with breast-conserving surgery, endocrine therapy and whole breast irradiation (BCS + ET + WBI) and surgery alone (BCS). This allowed multiple comparisons in a network meta-analysis.

The study endpoints were local recurrence (LR), regional recurrence (RR), distant metastasis-free interval (DMFI), disease-free survival (DFS), overall survival (OS), breast cancer-specific survival (BCSS), non-breast cancer death (NBCD), contralateral breast cancer (CBC), mastectomy-free interval (MFI) and secondary non-breast cancer (SNBC). LR was defined according to study protocols, including invasive as well as non-invasive ipsilateral breast cancer recurrence in four studies [13,21,22,23] and analyzed as the first event according to the included publications. One trial pooled local and regional recurrences [24,25] which were included in the local recurrence endpoint. DFS included any first local, regional or distant recurrence, contralateral breast cancer, secondary cancer and death without recurrence.

Because the results of the mastectomy rates in the NSABP B-21 trial were reported to be not statistically different, we assumed an equal distribution over the treatment groups [22].

Additionally, we also pooled the published individual patient data from the EBCTCG meta-analyses for the available endpoints BCSS and OS for the three trials in the direct comparison of BCS + WBI and BCS + ET [1,2,21,22,26,27]. Due to differences in the definition of the endpoint, any recurrences (including local and distant events) were not evaluable.

## 3. Statistical Analysis

Analysis of the studies includes patients’ characteristics as well as a description of the endpoints. Study-relevant events were extracted from the available publications, and hazard ratios, as well as odds ratios, were chosen as the appropriate comparison. Events were extracted as either the first or any event. Meta-analysis of the hazard ratios and odds ratios was performed using the inverse variance heterogeneity model. Statistical significance was set at a level of 95% resulting in a two-sided *p*-value of 0.05.

The Microsoft Excel plug-in MetaXl V5.3 (EpiGear International, Sunrise Beach, Australia) was used to analyze and pool the data. The figures were created using Microsoft Excel for Microsoft Office 365 Pro Plus (Redmond, WA, USA). Due to the possible heterogeneity of the study populations, the inverse variances of the heterogeneity model (ivhet) by Doi et al. were chosen as the comparison method [28]. This method favors larger trials, uses a more conservative estimation of the confidence limits and produces lesser-observed variances compared to the random effects model. Zero event correction was applied where appropriate [19]. Heterogeneity in the network was analyzed using H consistency [29]. For the network meta-analysis, we used the treatment of BCS + ET as the comparator arm as it represents the standard therapy in many current trial protocols. Heterogeneity within the meta-analysis was obtained with Cochran’s Q-test with the corresponding *p*-values. The search protocol was registered in the PROSPERO database with ID 418361.

## 4. Results

The results of the systematic literature review are shown in Figure A1. Table 1 demonstrates an overview of the included trials. We identified three trials that met the inclusion criteria for direct comparison (*n* = 1059 patients) [21,22,26,27]. For the network meta-analysis, we found ten trials, with seven comparing two treatment arms and two trials randomizing patients to three arms as well as one trial including four therapeutic arms (*n* = 7207 patients). The resulting network is shown in Figure 1.

The SweBCG91RT trial was included in a modified cohort. In this analysis, only HR+ Her2− patients without any adjuvant systemic therapy were included [30].

The median follow-up of the included trials was between 5.0–15.6 years, including low-risk tumors with mainly tamoxifen as endocrine therapy. In all network analyses, all H values were below 3, showing minimal network inconsistency.

Funnel plots for the direct analysis did not show any publication bias.

The analysis of the endpoint local recurrence is shown in Figure 2. The direct comparison between BCS + WBI and BCS + ET does not yield a significant difference (OR = 0.63 CI-95%: 0.34–1.16; *p* = 0.137). The indirect comparisons within the network analysis show a significantly better local control with BCS + WBI (HR = 0.62 CI-95%: 0.42–0.92; *p* = 0.019). Within the network, the addition of WBI to BCS + ET results in a significant reduction in LR (HR = 0.18; OR = 0.25; both *p* < 0.001). The omission of ET leads to a higher number of LR (HR = 1.95; n.s.; OR = 3.16; *p* < 0.001).

According to Figure 3, BCS + WBI and BCS + ET results in similar disease-free survival (direct: OR = 0.89; CI-95%: 0.55–1.44; *p* = 0.634). Within the network analysis, the trimodal therapy (BCS + ET + WBI) improves disease-free survival (HR = 0.67; OR = 0.70; both *p* < 0.001). Compared with BCS + ET, the omission of ET results in a significant reduction of DFS (HR = 1.97; OR = 3.64).

The direct comparison of overall survival between BCS + WBI and BCS + ET shows no statistically significant difference (OR = 0.93, CI-95%: 0.53–1.62, *p* = 0.785) (Figure 4). Similar results were obtained in the network analysis. The addition of WBI to BCS + ET does not result in a superior OS. The omission of ET leads to lower OS rates (OR = 2.50, CI-95%: 1.76–3.55, *p* < 0.001).

Table 2 shows the direct and network analyses for the three comparisons (BCS + ET + WBI vs. BCS + ET; BCS + WBI vs. BCS + ET; BCS vs. BCS + ET) for multiple additional endpoints (RR, DMFI, BCSS, NBCD, SNBC, CBC, MFI).

We observed significant differences in regional recurrences with the addition of WBI to BCS + ET (OR = 0.45, CI-95%: 0.24–0.83, *p* = 0.011). Distant metastases are statistically more likely in the BCS + WBI arm in the network comparison (OR = 2.10, CI-95%: 1.25–3.51, *p* = 0.005). These differences are not evident in the direct comparison. BCSS is lower after BCS alone when ET is omitted (OR = 4.49, CI-95%: 2.05–9.86, *p* < 0.001). BCS + WBI, in contrast to BCS + ET, results in a lower risk of dying for other reasons than breast cancer (OR = 0.61, CI-95%: 0.40–0.92, *p* = 0.020) in the network analysis. This observation was not seen in the direct comparison. In the treatment arms without ET, we observe significantly more contralateral breast cancers. Other secondary cancers are not significantly different in all comparisons. The mastectomy-free interval is improved by the addition of WBI compared with BCS + ET.

The comparisons of adjuvant WBI compared to adjuvant ET from the individual patient meta-analysis published by the EBCTCG (Figure 5) shows no difference in overall survival after 10 years of follow-up (direct: OR = 0.93, CI-95%: 0.60–1.44, *p* = 0.735; indirect: OR = 0.70, CI: 0.36–1.40, *p* = 0.315; combined: OR = 0.85, CI-95%: 0.59–1.22, *p* = 0.369). Likewise, breast cancer death also does not significantly differ between the two treatments (direct: OR = 0.62, CI-95%: 0.30–1.29, *p* = 0.202; indirect: OR = 1.10, CI-95%: 0.31–3.91, *p* = 0.879; combined: OR = 0.72, CI-95%: 0.38–1.36, *p* = 0.305).

## 5. Discussion

The oncological results of randomized trials assessing whole breast irradiation and endocrine therapy after breast-conserving surgery show in the direct and network comparison that both treatment options provide equally effective de-escalation strategies for women with low-risk breast cancer. The addition of WBI in the treatment paradigm improved local control and reduced the need for subsequent mastectomy after local recurrence. Non-breast cancer deaths might also be lower after BCS + WBI compared with BCS + ET. However, contralateral breast tumor recurrences were higher when omitting ET. The combination therapy of surgery, WBI and ET resulted in superior outcomes in LR and DFS but not OS or BCSS.

The results of this meta-analysis are mirrored by multiple databases and retrospective institutional analyses. These trials unanimously show no differences in both de-escalation strategies in terms of survival with favorable tendencies of local control with BCS + RT [36,37,38,39,40,41,42]. The choice of one therapy over another has to account for the very different toxicity profiles and application schedules. Endocrine therapy is currently applied using tamoxifen and/or aromatase inhibitors for a minimum time of five years using daily oral medications. The possible side effects of tamoxifen include increased risks for venous thromboembolism, uterine cancers, cataracts and fatty liver disease [2,43,44,45]. Further, AI has been shown to be more efficacious than tamoxifen in reducing recurrences and improving survival [3]. However, AIs are also associated with adverse events, such as a higher risk of osteoporosis, fractures, cardiovascular disease, diabetes and hypercholesterolemia. AIs were also linked to musculoskeletal pains and stiffness [46,47,48,49]. Both options for endocrine treatment are linked to hot flashes, sexual dysfunction, hair thinning and cognitive problems, including fatigue, forgetfulness as well as sleep disturbance [50,51,52]. However, an overall detrimental impact of ET on quality of life has not been consistently reported [52,53,54,55,56].

On the other hand, possible adverse events from whole breast radiotherapy include acute skin toxicity and fatigue as well as late toxicity with the risk of subcutaneous fibrosis, breast edema, breast pain, telangiectasia and secondary cancers [4,57,58,59,60]. Whole breast radiotherapy also has a small measurable impact on breast-specific quality of life. During the first three years of follow-up, women reported more breast symptoms. After year three, this difference was no longer present [10,34].

Due to limited adverse event data in the trials directly comparing WBI and ET, a formal analysis of adverse events could not be performed. The authors reported hot flashes, deep vein thrombosis and pulmonary embolisms associated with tamoxifen [22]. Changing the endocrine therapy from tamoxifen to aromatase inhibitors and the radiotherapy from whole to partial breast treatment with shorter schedules might change the efficacy and toxicity comparison.

Given the similar efficacy regarding DFS and OS, a detailed analysis to identify subgroups that might benefit from WBI or ET would be highly desirable. Unfortunately, information on specific patient groups was only available in the NSAPB B21 trial. In this analysis, the comparison of BCS + RT vs. BCS + ET resulted in statistically superior efficacy in local control for the age group 60–69 (HR = 0.31) and regardless of estrogen receptor status (HR = 0.31 and HR = 0.41) after WBI [22].

Radiation therapy schedules and treatment volumes have also changed considerably since the time when the included trials were conducted. First, current radiation schedules are shorter, with the majority of women treated with hypofractionated schedules consisting of 15–16 daily fractions. For these schedules, there is a high degree of certainty that they are equieffective with associated lower risks for acute and late adverse events [57,61,62]. More recently, the treatment schedules were even shortened further with the publication of the FAST-Forward trial using just five daily fractions for WBI [58]. Second, over the past two decades, multiple randomized trials have been conducted comparing whole breast radiotherapy to partial breast irradiation [63,64,65,66,67,68]. Despite some inconsistencies regarding different radiation techniques and fractionation schedules used for partial breast radiotherapy, the reduction of the treated breast volume has been shown to result in a significant improvement in acute toxicities as well as favorable cosmetic results [64,69,70,71,72]. Some schedules even allow for further treatment time reduction with lesser side effects and better QoL [63,72]. Generally, reducing the risk of local recurrence and forgoing salvage therapy is a valued objective for many patients leading to the majority preferring WBI to RT omission [73,74].

A surprising result in this analysis is the observation that patients undergoing WBI compared with ET alone had higher rates of distant metastases. When the trials were separately analyzed by the method of how the distant relapse events were scored (first, any, unknown), we observed higher DM rates only in the trials that reported DMs as first events. Given that WBI reduces local recurrences as the most common disease event, distant events would not be counted in patients that already suffered local relapses [1]. This would lead to a statistical artifact without clinical applicability.

The observed higher incidence of CBCs leads to the question of whether lack of endocrine therapy or the addition of WBI results in increased risk. As we detected the increase in the comparison of BCS + WBI to BCS + ET and not in the comparison of BCS + ET + WBI vs. BCS + ET, our results indicate the conclusion that the lack of ET and not the addition of WBI is mainly responsible for the increase in CBC.

The subsequent costs for the patients, health care system and providers are also important to consider when comparing different adjuvant treatment options. Multiple analyses demonstrated that radiotherapy was cost-effective in comparison to sole endocrine therapy after BCS [75,76,77]. Healthcare providers counseling patients on the appropriate de-escalation strategy might also consider that adherence to an adjuvant endocrine therapy also influences treatment outcomes. The number of women taking their medication over the full prescription time ranges between 50% and 85% [78,79]. Despite the fact that the adherence was probably not perfect in the analyzed trials, retrospective analyses suggest that poor adherence was associated with worse outcomes. Further, women that chose endocrine therapy alone were more likely to forgo the complete ET period with the risk of a higher relapse rate [80,81,82,83].

Limitations of the network part of the meta-analysis include that the comparisons are not based on individual patient data. However, trial-based analyses have also been demonstrated to provide equal results, which is also shown in our analysis in the comparison of OS and BCSS [84]. Here, the analysis based on trial data as well as individual patient data showed similar results.

Given the timeframe when the trials were conducted, the estimation of low risk was based on clinical features such as tumor stage and grading. Not all trials obtained information on the hormone receptor and Her2 receptor status [22], which might underestimate the effect of ET.

## 6. Future Directions

As mentioned before, multiple efforts are currently ongoing to identify a subgroup of patients who can safely omit WBI [14,15,16,17,18]. These inclusion criteria are based on different genetic essays trying to identify favorable prognostic groups with a minimal additive value of radiotherapy [85]. The inclusion of molecular classifiers in retrospective publications and as well as re-analyses of randomized data suggested that recurrence scores below 11, as well as PAM-50 scores, might be of value for the selection of RT omission [30,86,87,88,89,90]. The most recent POLAR score might even be of prognostic value [30]. However, these molecular tests should be evaluated in a prospective randomized trial before they are used in routine clinical practice. Currently, accruing prospective trials are also challenging the necessity of endocrine therapy in low-risk breast cancer. The EPOPE randomizes patients to accelerated partial breast radiotherapy using brachytherapy technique with or without ET [91]. The most intriguing study in this area appears to be the EUROPA trial asking whether BCS + PBI and BCS + ET result in similar QoL and local control [92]. Given the lack of oncological differences, patient-reported outcomes are especially important in this research area. Interestingly, in a patient survey, women reported that ET had the biggest negative impact on their QoL and would rather receive RT compared with ET [93]. More specific findings regarding patient-reported outcomes and adverse side effects may improve the integration of the patient perspective into the evaluation of different treatment types for early-stage breast cancer.

## 7. Conclusions

Based on the direct meta-analysis of three randomized trials as well as a network comparison, breast-conserving surgery with whole breast radiotherapy or endocrine therapy are equally effective de-escalation strategies in low-risk breast cancer in terms of disease-free and overall survival.

## Figures and Tables

**Figure 1 cancers-15-04343-f001:**
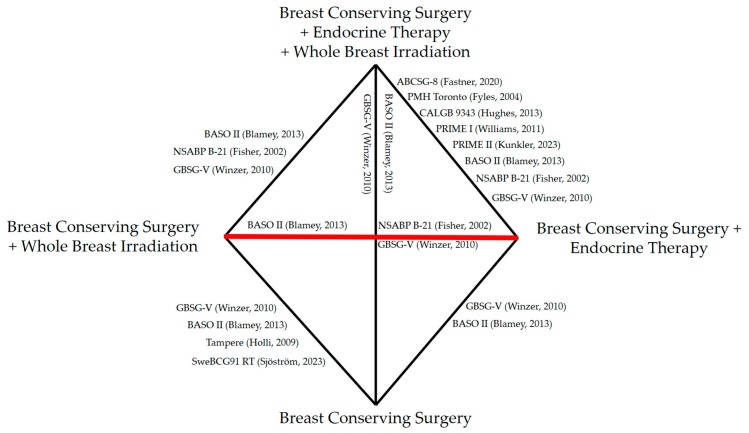
Overview of the analyzed network with the respective trials. Indirect comparisons are shown in black lines, and the direct comparisons are in red [5,6,10,21,22,24,27,30,31,32].

**Figure 2 cancers-15-04343-f002:**
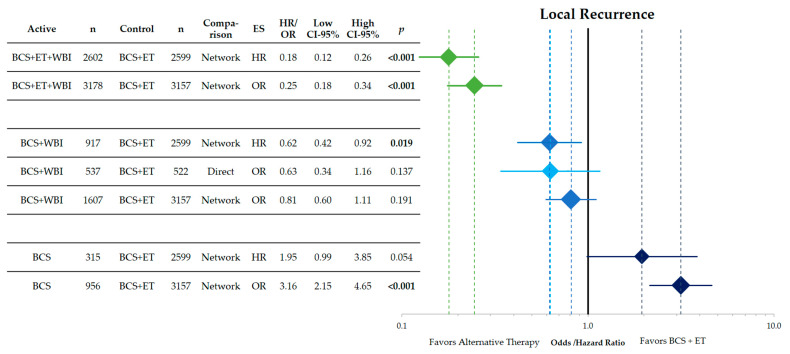
Forest plot of the network comparison of the endpoint local recurrence of different therapeutic approaches in low-risk breast cancer against breast-conserving surgery and adjuvant endocrine therapy. Shown are hazard and odds ratios with their corresponding 95% confidence intervals. The comparisons from top to bottom are BCS + ET + WBI vs. BCS + ET, BCS + WBI vs. BCS + ET and BCS vs. BCS + ET. The direct comparison of BCS + WBI vs. BCS + ET is shown in light blue. The width and height of the diamonds corresponds to the confidence interval. The dashed lines indicate the point estimates for each comparison. HR = hazard ratio, OR = odds ratio, CI = confidence interval, BCS = breast-conserving surgery, ET = endocrine therapy, WBI = whole breast irradiation, ES = effect size, *n* = number of patients.

**Figure 3 cancers-15-04343-f003:**
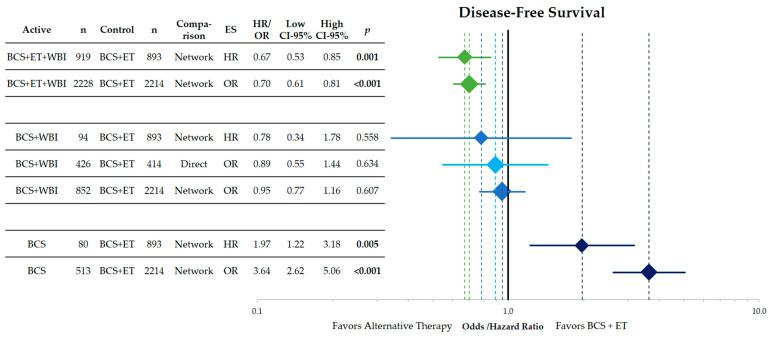
Forest plot of the network comparison of the endpoint of disease-free survival of different therapeutic approaches in low-risk breast cancer against breast-conserving surgery and adjuvant endocrine therapy. Shown are hazard and odds ratios with their corresponding 95% confidence intervals. The comparisons from top to bottom are BCS + ET + WBI vs. BCS + ET, BCS + WBI vs. BCS + ET and BCS vs. BCS + ET. The direct comparison of BCS + WBI vs. BCS + ET is shown in light blue. The width and height of the diamonds corresponds to the confidence interval. The dashed lines indicate the point estimates for each comparison. HR = hazard ratio, OR = odds ratio, CI = confidence interval, BCS = breast-conserving surgery, ET = endocrine therapy, WBI = whole breast irradiation, ES = effect size, *n* = number of patients.

**Figure 4 cancers-15-04343-f004:**
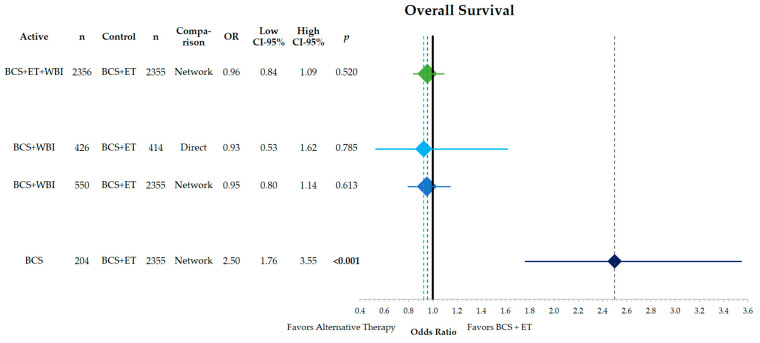
Forest plot of the network comparison of the endpoint of disease-free survival of different therapeutic approaches in low-risk breast cancer against breast-conserving surgery and adjuvant endocrine therapy. Shown are odds ratios with their corresponding 95% confidence intervals. The comparisons from top to bottom are BCS + ET + WBI vs. BCS + ET, BCS + WBI vs. BCS + ET and BCS vs. BCS + ET. The direct comparison of BCS + WBI vs. BCS + ET is shown in light blue. The width and height of the diamonds corresponds to the confidence interval. The dashed lines indicate the point estimates for each comparison. OR = odds ratio, CI = confidence interval, BCS = breast-conserving surgery, ET = endocrine therapy, WBI = whole breast irradiation, ES = effect size, *n* = number of patients.

**Figure 5 cancers-15-04343-f005:**
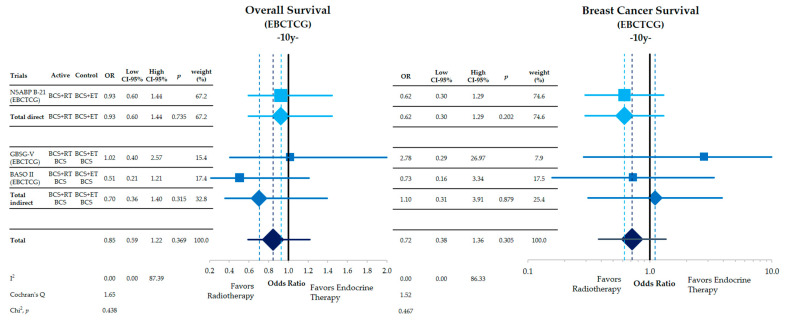
Comparative effectiveness of whole-breast radiation and endocrine therapy after breast-conversing surgery in low-risk women of overall survival and breast cancer-specific survival in the individual patient data meta-analysis from the EBCTCG after 10 y follow-up. Shown are the odds ratios with their corresponding 95-% confidence intervals. For one trial, the numbers and events are reported for the trial arms receiving BCS + RT and BCS + ET. For two trials, only pooled data together with the BCS-only arms are reported. This comparison is therefore termed “indirect”. The width and height of the diamonds and squares corresponds to the confidence interval. The dashed lines indicate the point estimates for each comparison.

**Table 1 cancers-15-04343-t001:** Overview of the patient characteristics of the included trials in the network meta-analysis.

Trial	Publications	Years Trial	*n* Total	FU [y]	Prim. EP	Inclusion	Strat.	Surgery	Axillary Staging	Systemic Therapy	Radiation Therapy	HR+	Treatment Arm	Control Arm
ABCSG-8	Fastner 2020 [31] Pötter 2007 [8]	1996–2004	869	9.9	LR	BCS, <3 cm, G1-2 ICD, G1-3 LC, N0, HR+	Age, Stage, Grade, Tam vs. AI, Center	Lumpectomy o. Wide Resection	SLNB/ ALND I-II	Tam or AI	40/2.66 Gy o. 50/2 Gy +opt. 10/2 Gy boost	>99%	BCS + ET + WBI	BCS + ET
PMH Toronto	Fyles 2004 [5] Fyles 2010 [33]	1992–2000	769	10	DFS	T1-2, cN0 o. pN0, Age > 50, R0	T-stage <2 cm, ER+-, Ax staging, Center	Lumpectomy	82% ALND	Tam 20 mg 5 y	40/2.5 Gy +12.5/2.5 Gy boost	94%	BCS + ET + WBI	BCS + ET
CALGB 9343	Hughes 2004 [7] Hughes 2013 [6]	1994–1999	636	12.6	LRR	T1, N0, cM0, Age > 70, ER+	Age >75 y, ALND	Lumpectomy	Clinical, ALND allowed	Tam 20 mg 5 y	45/1.8 Gy +14/2 Gy	78%	BCS + ET + WBI	BCS + ET
PRIME I	Prescott 2007 [34] Williams 2011 [10]	1999–2004	255	5	QoL	T0-2, N0, M0, >65 y	None	Lumpectomy	Sample, ALND I-III, SLNB	Tam 5 y	45–50/2–2.3 Gy +0–15 Gy Boost	n.r.	BCS + ET + WBI	BCS + ET
PRIME II	Kunkler 2015 [9] Kunkler 2021 [35] Kunkler 2023 [32]	2003–2009	1326	5	IBTR	T <= 3 cm, pN0, HR+, > 65 y	Center	Lumpectomy	Sample, SLNB, ALND	Tam 20 mg 5 y	40–50/2.0–2.66 Gy ggf. 10–15 Gy Boost	99%	BCS + ET + WBI	BCS + ET
BASO II	Blamey 2013 [21]	02/1992–10/2000	1135	10.1	LR	pT1, N0, G1 or spec. Histo, No LVI, <70 y	Unknown	Lumpectomy	Sample	Tam 20 mg 5 y	40/2.66 Gy o. 50/2 Gy +10–15/2–3 Gy Boost	n.r.	4 Arms: BCS vs. BCS + ET vs. BCS + WBI vs. BCS + ET + WBI	
NSABP B-21	Fisher 2002 [22]	1989–1994; 1996–1998	1009	8	IBTR	BCS T < 1 cm, Any Age	Age < >50 y	Lumpectomy	ALND I–II	Tam 10 mg BID 5 y	50/2 Gy +10/2 Gy boost	~57%	3 Arms: BCS + ET vs. BCS + WBI vs. BCS + ET + WBI	
GBSG-V	Winzer 2004 [26] Winzer 2010 [27]	1991–1998	347	10	DFS	pT1, pN0, 45–75 y, G1-2, L0, No EIC, HR+	Center	Lumpectomy	ALND I–II	Tam 30 mg 2 y	50/2 Gy +10–12/2 Gy Boost	~97%	4 Arms: BCS vs. BCS + ET vs. BCS + WBI vs. BCS + ET + WBI	
Tampere	Holli 2001 [25] Holli 2009 [24]	1990–1999	264	12.1	LRFS	Age > 40, ≤ T1, G1-2, Ki-67 < 10%	None	Sector Resection	ALND I–II	none	50/2 Gy	100%	BCS + WBI	BCS
SweBCG91 RT	Sjöström 2023 [30] Killander 2016 [13] Malmström 2003 [23]	1991–1997	597	15.6	IBTR	Age < 76 y, N0, Stage I-II, ER+, Her2−	Center, Detection	Sector Resection	ALND I–II	none	48–54/2 Gy No Boost	100%	BCS + WBI	BCS

**Table 2 cancers-15-04343-t002:** Overview of the direct and network comparison for multiple oncological endpoints using odds ratios and their respective 95% confidence intervals.

Active Therapy	*n*	Control Therapy	*n*	Comparison	OR	Low CI-95%	High CI-95%	*p*
Regional Recurrences								
BCS + ET + WBI	2230	BCS + ET	2217	Network	0.45	0.24	0.83	0.011
BCS + WBI	430	BCS + ET	417	Direct	0.92	0.10	8.95	0.946
BCS + WBI	430	BCS + ET	2217	Network	0.36	0.10	1.34	0.129
BCS	79	BCS + ET	2217	Network	0.60	0.08	4.52	0.617
Distant Metastases								
BCS + ET + WBI	2231	BCS + ET	1880	Network	1.15	0.75	1.75	0.522
BCS + WBI	430	BCS + ET	416	Direct	1.18	0.55	2.51	0.676
BCS + WBI	431	BCS + ET	1880	Network	2.10	1.25	3.51	0.005
BCS	204	BCS + ET	1880	Network	2.47	1.10	5.56	0.029
Breast Cancer-Specific Survival								
BCS + ET + WBI	2549	BCS + ET	2544	Network	0.74	0.49	1.10	0.137
BCS + WBI	430	BCS + ET	416	Direct	1.04	0.45	2.41	0.928
BCS + WBI	568	BCS + ET	2544	Network	1.44	0.86	2.41	0.163
BCS	204	BCS + ET	2544	Network	4.49	2.05	9.86	<0.001
Non-Breast Cancer Death								
BCS + ET + WBI	2546	BCS + ET	2542	Network	0.98	0.80	1.21	0.881
BCS + WBI	426	BCS + ET	414	Direct	0.86	0.28	2.71	0.802
BCS + WBI	564	BCS + ET	2542	Network	0.61	0.40	0.92	0.020
BCS	204	BCS + ET	2542	Network	0.87	0.46	1.66	0.678
Secondary Non-Breast Cancer								
BCS + ET + WBI	1472	BCS + ET	1465	Network	0.96	0.72	1.30	0.812
BCS + WBI	426	BCS + ET	414	Direct	0.94	0.35	2.54	0.906
BCS + WBI	426	BCS + ET	1465	Network	0.88	0.55	1.43	0.616
BCS	79	BCS + ET	1465	Network	0.82	0.37	1.80	0.613
Contralateral Breast Cancer								
BCS + ET + WBI	1886	BCS + ET	1882	Network	1.16	0.73	1.86	0.529
BCS + WBI	426	BCS + ET	414	Direct	2.78	1.19	6.52	0.019
BCS + WBI	564	BCS + ET	1882	Network	2.58	1.49	4.47	0.001
BCS	204	BCS + ET	1882	Network	3.31	1.14	9.59	0.028
Mastectomy								
BCS + ET + WBI	1312	BCS + ET	1324	Network	0.22	0.12	0.40	<0.001
BCS + WBI	336	BCS + ET	336	Direct	0.50	0.25	1.00	0.049
BCS + WBI	474	BCS + ET	1324	Network	0.56	0.30	1.06	0.076
BCS	125	BCS + ET	1324	Network	0.82	0.37	1.80	0.613

## Data Availability

Data used in this trial are available in the referenced publications.

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
