# Peer review of "Whole Breast Irradiation in Comparison to Endocrine Therapy in Early Stage Breast Cancer—A Direct and Network Meta-Analysis of Published Randomized Trials"

_cancers, 2023, doi:10.3390/cancers15174343_

Round 1

Reviewer 1 Report

I'd like to thank the authors for this interesting and informative meta-analysis comparing adjuvant treatment de-escalation in early stage BC (stages I and II, N0), omitting either ET or WBI in the sacrosanct tripod of BCS-WBI-ET.

By using the results of randomized trials (direct and indirect comparisons), they reported LR, LRR, DMFI, DFS, OS, BCSS, NBCD, CBC, MFI, and SNBC.

The main result of this analysis is that omitting either ET or WBI leads to equivalent OS and DFS.

However, LR and DFS are significantly improved by the addition of WBI to BCS + ET while OS is not.

The overview of the direct and network comparison adds information on the advantages and drawbacks of each de-escalation choice: while the tripod remains safer in terms of LR and DFS, choosing ET as the sole adjuvant treatment to BCS prevents CBC and possibly DM while choosing WBI increases the CBC rate but lowers the LR rate, non-breast cancer death (due to ET side effects ?) and increases the MFI.

This meta-analysis might prove helpful when counseling patients in treatment de-escalation, taking into account patients' convenience and possible side effects of each adjuvant treatment as well as the low observance of ET over the long term and its possible impact on patients' outcomes.

 I do not feel qualified to judge the statistical method.

I would just suggest that the authors incorporate in the discussion section a little more data about patients' willingness. I mean, in patients' choice studies investigating BC patients’ preferences for adjuvant radiation therapy, a low rate of patients opt for adjuvant radiation avoidance, meaning that for patients, LR matters. This remark would mitigate the statement that « both treatment options provide equally effective de-escalation strategies for women with low-risk breast cancer » that should be complemented by « in terms of DFS and OS ».

Reviewer 2 Report

The authors reported on whole breast irradiation in comparison to endocrine therapy in early stage breast cancer – a direct and network meta-analysis of published randomized trials. This a relevant and heavily discussed question in the field of adjuvant treatment of early breast cancer. It is of interest for the readers of Cancers.

The manuscript is well written, clear, relevant for the field and presented in a well-structured manner.

The cited references are mostly recent publications and relevant.

The manuscript is scientifically sound and the experimental design is appropriate to test the hypothesis.

Statistical analyses were well performed. Besides some typos and lacking explanations the figures and tables are appropriate.

The conclusions are consistent with the evidence and arguments presented.

Some typos need to be fixed.

Specific comments:

1. Discussion l235:

' contralateral breast tumor recurrences were higher when omitting ET' could mean a lack of  effectiveness without ET or an increased risk for a secondary contralateral breast cancer because of radiotherapy doses to the contralateral breast. If possible please comment on that.

2. Discussion l279: 

The sentence ' Yet, there is also a small increase in elsewhere recurrences in the ipsilateral breast with PBI which is why the use of PBI has been restricted to low risk patients [75].' is an unacceptable generalization of the efficacy of APBI. In your metaanalysis this higher LR-rate is related to intraoperative APBI techniques. This should be clarified.

btw: The majority of working groups did not restrict APBI to low risk patients because of higher ipsilateral local recurrence rates! Decades ago, these working groups found out that patient selection is a precondition for a successful APBI, other working groups especially those using intraoperative electron were not aware of this fact when designing their trial. 

Therefore this sentence should be clarified.

Minor editing of English language required:

Suggestions:

p2 l72: successfully

p2 l74: delete extra space

Table1: ABCSG-8 line: BCS+ET

Table1: PRIME II line: delete ggf., use opt. instead

Table1: Please explain all abbreviations under the table

Figure 4: Please delete extra space in the captions. 

p10 l289: anal-yses looks a bit strange, I have some doubts that this is the appropriate way to write, but I am not sure

Reviewer 3 Report

The purpose of this study is to compare the long-term results of radiation therapy to the whole breast versus endocrine therapy after breast-conserving surgery for early-stage breast cancer. In my opinion, this study is intriguing and appropriate for publication after some modifications and clarifications.

Abstract: line 40-41: acronyms used for the first time should be spelled out in full.

Introduction

-       I think it is important to clarify and scientifically support the need for therapeutic de-escalation by using only endocrine therapy versus whole breast radiotherapy. As for now, the standard of care is the combination of surgery, radiotherapy, and endocrine therapy. Why do we feel the need to de-escalate, and, more importantly, in which category of patients? This consideration should be included in both the introduction and the discussion.

-       When referring to low-risk patients (line 57), this category of patients should be better defined, e.g., "small primary tumor" What pT or diameter are we talking about? or "low proliferation index" refers to what cut-off? Etc.

Results: 

-       Table 1 must be implemented by making the inclusion criteria much clearer, I suggest making a column for each and indicating if the criterion was not considered in that study. Also, the treatment arm and control arm columns, while understandable, are not quick to use. Let the table be horizontal so that everything can be included. In the first line, there is probably a mistake in the acronyms, HT should be replaced by ET. I also suggest adding the acronyms to the table caption.

-       For clarity in reading the results, the order of presentation in the text should follow that of the figures, so unless you decide to change the figures, first the three combination therapies vs. BCS+ET, then BCS+WBI vs. BCS+ET, and lastly BCS vs ET

-       Line 207: “BCSS was lower when ET was omitted”: clarify that it is BCS versus BCS+ET.

Discussion: 

-       I reiterate that it is useful to reinforce the reason behind the need for this study, why and in whom we feel the need to de-escalate from what is the standard of care.

-       The paragraph from 273 to 280 seems off-topic to me.

-       from line 280: I do not find the result surprising, rather I would include this consideration within the limits of the study.

-       Line 302: first-time use of IPD, is to be spelled out in full.
